# Spatially Explicit Fuzzy Cognitive Mapping for Participatory Modeling of Stormwater Management

Corey T. White [1,*], Helena Mitasova [1,2], Todd K. BenDor [3], Kevin Foy [4], Okan Pala [1], Jelena Vukomanovic [1,5] and Ross K. Meentemeyer [1,6]

1   Center for Geospatial Analytics, North Carolina State University, Raleigh, NC 27695, USA;
    hmitaso@ncsu.edu (H.M.); opala@ncsu.edu (O.P.); jvukoma@ncsu.edu (J.V.); rkmeente@ncsu.edu (R.K.M.)
2   Marine, Earth and Atmospheric Sciences, North Carolina State University, Raleigh, NC 27695, USA
3   Department of City and Regional Planning, University of North Carolina at Chapel Hill,
    Chapel Hill, NC 27599, USA; bendor@unc.edu
4   School of Law, North Carolina Central University, Durham, NC 27707, USA; kfoy@nccu.edu
5   Parks, Recreation and Tourism Management, North Carolina State University, Raleigh, NC 27695, USA
6   Department of Forestry and Environmental Resources, North Carolina State University,
    Raleigh, NC 27695, USA
*   Correspondence: ctwhite@ncsu.edu

**Abstract:** Addressing "wicked" problems like urban stormwater management necessitates building shared understanding among diverse stakeholders with the influence to enact solutions cooperatively. Fuzzy cognitive maps (FCMs) are participatory modeling tools that enable diverse stakeholders to articulate the components of a socio-environmental system (SES) and describe their interactions. However, the spatial scale of an FCM is rarely explicitly considered, despite the influence of spatial scale on SES. We developed a technique to couple FCMs with spatially explicit survey data to connect stakeholder conceptualization of urban stormwater management at a regional scale with specific stormwater problems they identified. We used geospatial data and flooding simulation models to quantitatively evaluate stakeholders' descriptions of location-specific problems. We found that stakeholders used a wide variety of language to describe variables in their FCMs and that government and academic stakeholders used significantly different suites of variables. We also found that regional FCM did not downscale well to concerns at finer spatial scales; variables and causal relationships important at location-specific scales were often different or missing from the regional FCM. This study demonstrates the spatial framing of stormwater problems influences the perceived range of possible problems, barriers, and solutions through spatial cognitive filtering of the system's boundaries.

**Keywords:** flooding; geospatial analytics; GRASS GIS; knowledge elicitation; spatial scale dependency; socio-environmental systems; climate change; urban growth; socio-hydrology

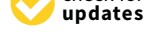

## 1. Introduction

In polycentric urban regions, disparate stormwater management authorities segment physical and political boundaries, creating a challenging management environment. As rapid urban development and climate change advance, flooding and water quality degradation are forcing stormwater managers, policy-makers, and communities to approach these challenges together at new scales. For example, in areas where water quality fails to comply with state or federal regulations, such as the US Clean Water Act, the cost of mitigation can be a significant burden on a city or town [1]. In addition, the costs of stormwater mitigation strategies can cause a public and political backlash against major improvements to the stormwater system at large, especially when coupled with ineffective management structures. With all of the factors involved, urban stormwater problems are considered "wicked" problems, where different stakeholder groups interpret problems differently and sometimes do not see the same problems at all [2]. The first step in addressing sustainability

problems is building a shared understanding of them among the stakeholders with the influence to develop solutions [3]. These stakeholders include "top-down" (i.e., scientist, policy-makers) and "bottom-up" (i.e., local managers, environmentalists) each of which work to solve these problems at different spatial scales [3]. Through the participatory modeling process these distinct perspectives are untangled and framed to provide a shared understanding of the problem to create sustainable adaptation measures [3,4].

Fuzzy cognitive maps (FCM) were developed by Bart Kosko [5] as an extension of cognitive maps, which are used to model people's perceptions of complex systems [3,6]. Kosko extended cognitive maps by introducing fuzzy logic [−1, 1] instead of crisp (binary) logic when modeling the relationships represented within the graph. He also introduced the idea of computationally modeling the effect of different policy options on the network [5,6]. FCMs are a semi-quantitative participatory modeling method for community-based modeling of socio-environmental systems (SES) [5–7]. FCMs provide a fast conceptual modeling solution that works with diverse stakeholder groups because they are simple to create and interpret [7,8]. However, FCMs have limitations when modeling SES, especially around considerations of space and time. For example, FCMs do not provide spatial-temporal characteristics of the modeled system, and they are limited in their ability to model time delays between causal relationships. Therefore, they only represent a snapshot of participating stakeholders' cognitive models [9,10]. Nevertheless, FCMs are a popular choice in participatory modeling, where a developed FCM itself can be the community goal or a stepping stone into a more complex participatory modeling method [9].

Although FCMs have historically not been spatially explicit, Gray et al. (2014) suggest that FCM generated at different spatial scales (i.e., extent and grain) contain different properties and that these scaling differences are a hurdle that need to be addressed to develop sustainable adaptation strategies to challenges in SES. However, Gray et al. (2014) do not elaborate on the analysis of how the stakeholder-derived spatial boundaries impacted the types of variables and connections made. System dynamic (SD) literature provides additional insights into the role of scale on causal relationships in the SES [11]. BenDor et al. (2012) demonstrate how representing space (i.e., field and object) and spatial scale can influence the model. However, little research has examined how the spatial scale of an FCM affects simulation of real-world processes for decision-making [12]. Furthermore, individuals' personal and professional experiences at particular locations and spatial scales implicitly affect the system boundaries of the FCMs they create. For instance, if two city planners from neighboring cities, who each have a detailed understanding of spatially explicit problems in their town, were asked to create FCMs representing affordable housing, each planner would implicitly base their FCM on their own understanding and likely the spatial scale of their own city. How these FCMs would scale up or down to represent larger or smaller spatial extents has not been studied. While FCMs provide conceptual (non-spatial) frameworks for understanding problems and their processes, spatially explicit surveys or participatory GIS (PGIS) studies ask stakeholders to describe the conditions at specific locations [13], eliciting local knowledge about natural resource and socio-environmental challenges.

This paper couples spatially explicit surveys and FCMs to (1) understand the role of spatial scale on the development of FCMs, (2) explore which spatial scales stakeholders identify for spatially explicit stormwater problems, and (3) examine if stakeholders accurately represent the spatial characteristics of the landscapes they are modeling. Linking conceptual understanding of system dynamics and simulating processes in actual locations is important for real-world problem solving. To demonstrate our approach, we worked with stakeholders from the Research Triangle region of North Carolina, where water quality has fallen below federal regulations [1]. For the City of Durham, the implementation of proposed stormwater regulations could have cost nearly $680 million in infrastructure improvements, potentially forcing the city into financial insolvency without outside financial assistance [1].

## 2. Methods

We divided the methods section into five distinct subsections: Section 2.1 introduces the study system, and Section 2.2 provides an overview of the participatory modeling workshops we held with stakeholders. In Section 2.3, we describe FCM model development, followed by FCM analysis (Section 2.3.1) and FCM model aggregation and scenario evaluation (Section 2.3.2). Next, Section 2.4 addresses spatially explicit surveys and the analysis of spatial characteristics of stormwater problems (Section 2.4.1). Finally, in Section 2.5, we introduce spatially filtered fuzzy cognitive maps by explaining how to couple FCMs with spatially explicit survey data.

### 2.1. Study System

Over a series of participatory modeling workshops with planning and policy professionals from academia, government, industry, and non-governmental organizations (NGOs), we co-developed a shared understanding of stormwater management in the Research Triangle metropolitan region of the State of North Carolina (USA). The region's core is formed by the City of Durham in Durham County, the Town of Chapel Hill in Orange County, and the state capital, the City of Raleigh in Wake County (Figure 1). The area's population grew 21% from 2010 to 2019 and is now nearly 1.6 million [14]. The region is rapidly developing over a complex hydrological landscape divided between the Neuse and Cape Fear River Basins. Wake (847 km$^2$), Durham (269 km$^2$), and Orange (142 km$^2$) Counties contain a total of 1258 km$^2$ developed land, with remaining land cover mostly made up of forested (1852 km$^2$) or planted/cultivated (537 km$^2$) land. Soils in the Triangle are mostly clay, which means that they have a low infiltration rate and high runoff potential [15]. The Triangle also experiences regular, severe thunderstorms over the summer months and frequently experiences direct and indirect effects from hurricanes (Hurricane Floyd 1999, Hurricane Matthew 2016, Hurricane Florence 2018). These weather conditions cause significant pluvial (i.e., pooling from stormwater runoff) and fluvial (i.e., overflown lake, river, or stream) flooding within the region.

Stormwater management in the Triangle is complicated by the wide variety of communities, organizations, and governing bodies in the region. As the Triangle continues to develop, urban stream channels are degrading water quality [16]. High nutrient loading and eutrophication from stormwater runoff are also harming aquatic ecosystems [16]. Multi-governmental collaboration and community support are needed to address these concerns because of the financial constraints of implementing solutions and the lack of jurisdictional power of any single municipality to define problems or enact solutions [16]. Additionally, the Triangle lacks incentives for collaboration among various governing and management organizations and instead relies on regulations from federal and state government bodies to enforce rules [16]. However, regional governments such as the Triangle J Council of Governments (TJCOG) are attempting to bridge these gaps by developing new collaborative programs (e.g., regional stormwater data sharing standards) [17].

### 2.2. Participatory Workshops

To build common understanding of stormwater management challenges in the Triangle, two workshops were held at the Center for Geospatial Analytics at North Carolina State University (Figure 2). Stakeholders were recruited with directed emails to individuals from academia, government, industry, and NGOs from the stormwater community in the Triangle. The first workshop was held as an initial fact-finding meeting with stakeholders in May 2018. The workshop consisted of 24 professionals from around the Triangle region, including local, state, and federal government employees, academics, and one individual from industry. During the workshop, participants were asked to complete surveys to identify which stormwater issues are facing the Triangle, the barriers to fixing these issues, and strategies that can be taken to overcome these barriers [16].

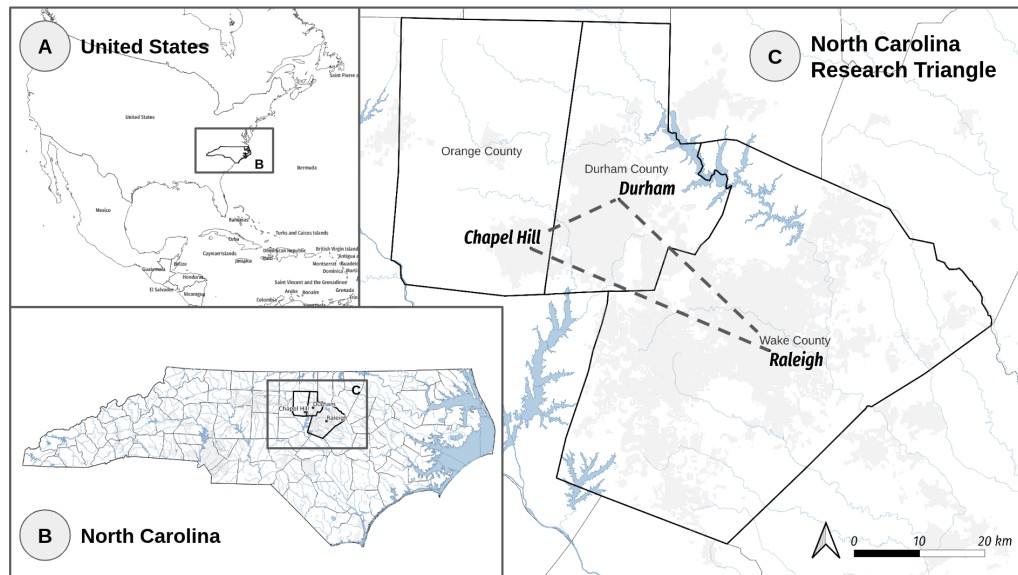

**Figure 1.** The North Carolina (**B**) Research Triangle (**C**) region (USA (**A**)) includes the Town of Chapel Hill, the City of Durham, and the City of Raleigh, and a number of other communities. The area has a rapidly growing population, now estimated at 1.6 million (2020).

The second workshop was held in April 2019 and included 25 stakeholders, of which six were participants in Workshop 1. The purpose of the second workshop was to co-develop a regional model of stormwater management (using data from Workshop 1 [16]) and to identify the locations of specific stormwater problems, including those where stakeholders disagreed on their cause and severity (i.e., wicked problems). During the workshop, FCMs were developed by individual stakeholders using MentalModeler (http://www.mentalmodeler.org, (accessed on 12 April 2019)), an online application designed to develop FCMs [18]. Stormwater problem areas were individually identified by stakeholders using the spatial survey application Survey123 for ArcGIS. Survey123 for ArcGIS is an online spatial surveying platform developed by Esri, which provides both web and mobile engagement interfaces (https://survey123.arcgis.com (accessed on 12 April 2019)).

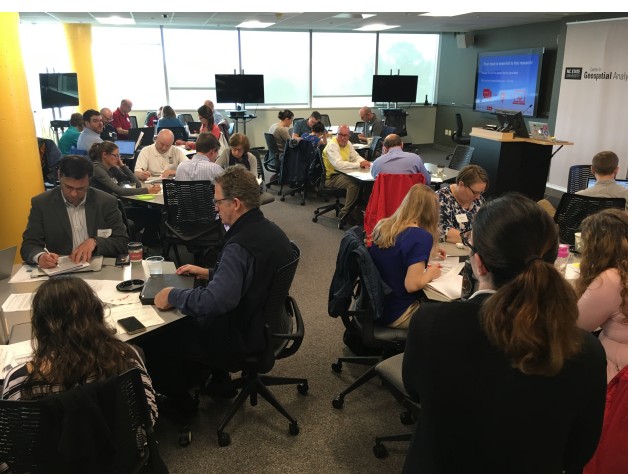

**Figure 2.** Stakeholders representing planning and policy professionals from across the Research Triangle, NC, participated in a workshop held at the Center for Geospatial Analytics at North Carolina State University to co-design stormwater management models.

### 2.3. Modeling Local Knowledge with Fuzzy Cognitive Maps

Fuzzy cognitive maps are represented as directed graphs that are composed of variables (N) and connections (C) assigned weights from the set [−1, 1] to define the causal relationships between the variables [5]. An FCM can be drawn as a graph or adjacency matrix $A(D) = [a_{ij}]$ [6,19] and can be evaluated using graph indices and similarity coefficients [5,6]. After an FCM is developed, the model can explore "what if" scenarios by activating variables in the model. A variable is activated by setting its activation state to a value in the set [−1, 1]. These activation values can be thought of as relative amounts such as "a lot" or "a little". An example scenario would be asking the question, "what if we reduce impervious surfaces a lot?" Running this scenario would involve setting the variable "impervious surfaces" to an activation state of −1. The model would then show the relative effects of "decreasing impervious surfaces a lot" on the remainder of the system.

During Workshop 2, stakeholders completed a short training period where they were taught how to develop an FCM as a group. After training, the stakeholders were given one hour to individually create an FCM on their personal or a provided laptop using the following prompt:

*"When I mention stormwater management in the Triangle what variables/things come to mind? How do these things affect each other?"*

After each stakeholder completed their FCM, they were instructed to export their model and upload the results to a shared Google Drive to be further analyzed using code from the open-source Python library PyFCM [20].

### 2.3.1. Analysis of Stakeholder Groups' FCMs

To determine if the sample size of FCMs created during the workshops was large enough to capture the complexity of the modeled system, accumulation curves were computed using Monte Carlo techniques to generate 500 possible accumulations of the FCM model components [6]. The accumulation curves represent the cumulative aggregation of unique variables and the decay of unique variables with each additional FCM. The accumulation curves will flatten at a stable state when each additional FCM adds few to no new variables to the system [6].

Graph measures and indices were calculated for each FCM to compare the structure of the stakeholders' FCMs individually and as stakeholder groups. The measures include the number of variables (N), connections (C), transmitters (T), receivers (R), ordinary variables, and the in-degree and out-degree of each variable (explained below) [6]. The graph indices include complexity (R/T), density (D), and hierarchy (h) [6].

The in-degree is the absolute sum of a variable's column in the adjacency matrix and represents how much the variable is influenced by its inward connection, while the out-degree is the absolute sum of a variable's row in the adjacency matrix that represents how much cumulative influence it has on other variables [6]. Centrality is the sum of the in-degree and out-degree and represents the cumulative influence the variable has on both its inward and outward connections [5,6].

A transmitter variable is a forcing function in a cognitive map where a variable only contains outward facing connections and an in-degree of zero, while a receiver variable represents utility functions that contain a positive in-degree and an out-degree of zero [6]. Variables that have both positive in-degree and out-degree are considered ordinary variables [6,21]. Complexity is the ratio of receivers (R) over the number of transmitters (T) in an FCM. When an FCM has a large number of receivers (R), that indicates that the stakeholder envisions many possible outcomes within the system. A large number of transmitters (T) indicates that the model has many system drivers [6]. An FCM with a large ratio has more complexity because there are many possible outcomes within a system with few forcing functions [6].

Similarly, density (D) indicates the connectedness (or sparseness) of casual relationships in an FCM [6]. FCMs with greater density contain more causal relationships and

present more options for intervention [3,6], while FCMs with low densities may represent that stakeholders perceive fewer management options [3,6].

The hierarchy index (h) ranges from zero to one; an FCM with a hierarchy of one is considered hierarchical, and an FCM with a value of zero is democratic [6,22]. A democratic FCM (low hierarchy) indicates that the stakeholder is more likely to believe the system can adapt to local SES changes and may represent good cognitive models to use as a starting point for management strategies [6,23]. A hierarchical FCM (high hierarchy) indicates that the stakeholder considers the system from the top-down (system drivers), thus having less belief that the system is easily changed [6,23].

We statistically compared the FCMs' indices using either ANOVA [24] or Mann–Whitney after each sample was tested for normality using the Shapiro–Wilk test of Normality [6]. ANOVA was used when the Shapiro-Wilk test determined the data was normal while Mann–Whitney was used when data normality was rejected [6].

Additionally, we compared the FCMs' variables by analyzing similarity coefficients of the variables, such as the most mentioned variables, most frequent transmitters and receivers, and most frequent central variables [6]. We used Pearson's Chi-Square ($\alpha \leq 0.05$) to examine if stakeholder groups differed significantly in their selection of variables to create an FCM [25], and the Jaccard similarity coefficient ($SJ(A, B)$) was used to compare the similarity between two FCMs' variables [26,27]. The Jaccard similarity coefficient works by comparing two sets; Set $A \in \{a_1, a_2, a_3, ...\}$ and set $B \in \{b_1, b_2, b_3, ...\}$ where each set contains the variables of a distinct FCM. The similarity coefficient is then calculated by dividing the total number of variables found in the intersection of sets A and B by the union of the total variables from sets A and B.

$$SJ(A, B) = \frac{|A \cap B|}{|A \cup B|} \tag{1}$$

### 2.3.2. Model Aggregation and Evaluation of Scenarios

Social cognitive models represent a collection of individual FCMs that are combined using qualitative or quantitative aggregation methods [6]. From the individual FCMs, we generated a social cognitive model, that we call the 'Regional FCM' (RFCM, $E_c$), using a mixed method where the individual FCMs' variables were first qualitatively coded before quantitative aggregation. Two members of the research team inductively coded the variables found in the individual stakeholders' FCMs on two separate occasions. During this process, similar variables were subjectively combined into new groupings. The coders met to compare their efforts in both cases to establish rules for coding and to compare coding results. After all discrepancies were rectified, a final coded set of variables was produced. Variables occurring in only one FCM were removed to reduce noise and simplify the model. Although this action dramatically decreased the complexity of the social cognitive model, we acknowledge that it will inherently be a conservative representation of the stakeholders' mental models, and may, in certain edge cases, poorly represent minority opinions [6]. Next, we generated an adjacency matrix $E_i$ for each stakeholder's FCM containing the distinct variables from all of the FCMs, while maintaining the original connection values. Finally, the augmented adjacency matrices were aggregated using matrix algebra to calculate the mean weight of each variable, where *k* is the number of stakeholders who included the coded variable [28,29]. The resulting adjacency matrix ($E_c$) represented all of the stakeholders' FCMs combined into a single FCM that we call the regional FCM (i.e., social cognitive model or aggregated FCM).

$$E_c = \frac{1}{k} \sum_{i=1}^{k} E_i \tag{2}$$

FCMs allow stakeholders to run scenarios that simulate possible management or policy options in a system [28]. We ran scenarios with the sigmoid activation function to compare FCMs generated at different spatial scales [8,30,31]. When comparing model

outputs, we were not concerned with the absolute values of the results, but instead focused on how variables were changed under each scenario, and their directions of change.

### 2.4. A Spatial Survey to Identify "Wicked" Stormwater Problems

Facilitating the collection of crowd-sourced geospatial data allows individuals to supply local knowledge about areas where they have personal experience [32–35]. Web-based, spatially explicit surveys are commonly used to crowd-source geospatial data about SES because they increase levels of participation by providing an accessible engagement platform [33,36,37]. The power of spatially explicit engagement platforms lies in stakeholders being able to feel ownership of an issue through personal connection to place [32]. Spatially explicit surveys also provide a method to elicit stakeholder knowledge about the spatial scale of an issue, which can often be overlooked [32,36]. During Workshop 2, stakeholders were asked to complete a spatially explicit survey using Survey123 for ArcGIS to identify the geographic locations of stormwater problems in the Triangle by placing a point on the map (Link to survey: https://arcg.is/14O990, (accessed on 23 February 2021)). Specifically, they were asked to consider the following questions about ongoing and future stormwater problems in the Triangle, NC:

- Where are some stormwater problems that currently exist in the Triangle, NC? What about in the near future?
- What are some barriers that are preventing or may happen to prevent these problems from being managed?
- What kind of actions need to be taken to resolve these problems?

Each stakeholder was asked to complete this task individually for as many locations as they would like. The survey was designed to locate and determine the spatial-temporal scale of stormwater problems, barriers, and solutions by asking stakeholders to identify the spatial and temporal scales of the drivers and impacts of the problem.

Spatial scale ($Ss$) is the 2-dimensional combination of spatial extent ($Se$) and spatial grain, where extent represents the area of the problem and spatial grain ($Sg$) is the level of detail of the impacted process [38].

$$Ss = Se/Sg \tag{3}$$

For each location they identified, stakeholders were asked the following questions and could select either household, neighborhood, city, regional, global, or not sure:

- At what spatial scale does the driver of this problem occur?
- At what spatial scale does the impact of this problem occur?

The temporal scale ($Ts$) is represented by the temporal extent ($Te$) or how often the problem occurs over the temporal grain ($Tg$), the unit in which the impact process is measured [38].

$$Ts = Te/Tg \tag{4}$$

Stakeholders were asked to select either day, week, month, year, or decade when responding to the following questions:

- How often does the problem occur?
- How long does the impact of the problem last?

The survey results were then analyzed and compared to the spatial characteristics of the described locations.

### 2.4.1. Analysis of Stormwater Problem Clusters

Stormwater problem hotspots were identified by clustering the data collected from the spatially explicit survey using the GRASS GIS [39] *v.cluster* module. Density-Based Spatial Clustering of Applications with Noise (DBSCAN) [40] was the chosen clustering

algorithm, and it was parameterized so that clusters must contain a minimum of three features with a maximum search distance of 1000m. The clusters represent the areas that had the most consensus among stakeholders about the location of stormwater problems. From these clusters, we examined how stakeholder definitions of the spatial-temporal scale and the spatial characteristics of the areas compared to existing geospatial data representing the landscape.

To begin the analysis, we positioned each point in a cluster to its nearest stream segment before calculating the upstream contributing area for the cluster using the GRASS GIS plugin *r.stream.snap* and *r.stream.basin* [41]. We used the upstream contributing area of the cluster to represent the hydrological spatial extent of the problem, which we then compared to the stakeholder-defined spatial scale of the driver. We then evaluated land cover, development patterns, and simulations of surface water runoff and flood inundation within each cluster's upstream contributing area to compare how stakeholder knowledge aligned with existing geospatial data sources.

The land cover characteristics of each cluster's upstream contributing area were derived from the National Land Cover Database 2016 [42]. Developed land was represented by NLCD classes 21 thru 24: Developed (Open Space), Developed (Low Intensity), Developed (Medium Intensity), and Developed (High Intensity). We compared stakeholder descriptions of the total area of developed land to the NLCD data for each cluster.

To explore how the development in each cluster's upstream contributing area changed over time, we used building footprint data to generate a map depicting the decade in which each building was developed. To do this, we converted the building footprint data to centerpoints, where each point contained a value representing the year the building was built. The centerpoints were then interpolated using the regularized spline with tension method implemented in GRASS GIS as the module *v.surf.rst* [43]. Finally, we removed undeveloped land and water from the surface by masking the surface to only include areas overlapping with NLCD 2016 developed classes.

Floodplain data was broken into three sections composed of the floodway, 1% chance of flooding (Prior 100yr floodplain), and 0.2% chance of annual flood (Prior 500 year floodplain). To validate stakeholder knowledge about the current amount of development within the floodplain, we used the FEMA floodplain boundary data from Orange and Wake counties to calculate the total area of developed land (NLCD) within the floodplain. We also validated that specific buildings mentioned by stakeholders were located within the floodplain by using building footprint data provided by the State of North Carolina.

However, floodplain data underestimates actual flooding hazards in an area, especially pluvial flooding [44]. To account for this, we simulated pluvial flooding for each cluster's upstream contributing area, using the process-based geospatial stormwater runoff model SIMWE (SIMulated Water Erosion) implemented in GRASS GIS as the module *r.sim.water* [39,45]. SIMWE simulates overland flow by using Green's function Monte Carlo method to solve the 2D shallow water flow described by the bivariate form of Saint Venant equations [45]. The model allows the use of variable terrain, land cover, soil, and rainfall excess conditions [45,46]. A 10-m DEM (QL2 LiDAR) from the NC Division of Emergency Management (https://sdd.nc.gov/ (accessed on 23 February 2021) was used as input elevation data to the model. Manning's roughness coefficient was calculated from NLCD 2016 land cover data to provide variable friction [47]. The simulation was parameterized to run for one hour with a rainfall intensity of 79.248 mm/h in each of the clusters upstream contributing areas in order to represent a 100-year precipitation event for the region [48].

We also simulated a one-meter maximum inundation event inside each cluster's upstream contributing area to compare stakeholders' descriptions of flooding and the spatial scale of the impact of flooding. To calculate inundation, we used the height above nearest drainage (HAND) method using GRASS GIS [39,49]. From the calculated inundation data, we were able to determine maximum flood depths and total flooded area.

### 2.5. Spatially Filtered Fuzzy Cognitive Map

Fuzzy cognitive maps contain two distinct types of spatial scale, the spatial frame and the system dynamic scales (Figure 3). The spatial frame of the FCM is the spatial scale used to prompt (e.g., bound, frame) the development of the FCM. The spatial extent of the spatial frame is represented by the area of interest (e.g., city, watershed, etc.) [3]. The spatial grain is the level of detail considered when the variables and relationships are considered. Thus, the spatial extent implicitly sets the spatial grain of the spatial frame. For example, if an FCM is created for a river basin, the variables and relationships considered will differ from an FCM developed for a subwatershed. The system dynamic scales are the spatial scale of an FCM represented by a multi-scale system, where causal relationships define the spatial scale of the variables. The multi-scale system is the system boundary of the FCM, which is made up of the intersection of systems that form the model. For example, stormwater management is a subsystem of many systems containing variables from both societal and environmental systems. The spatial scale of each variable in a system is dependent on the context of its relationships. However, a variable without the context of its relationships maintains an independent spatial scale.

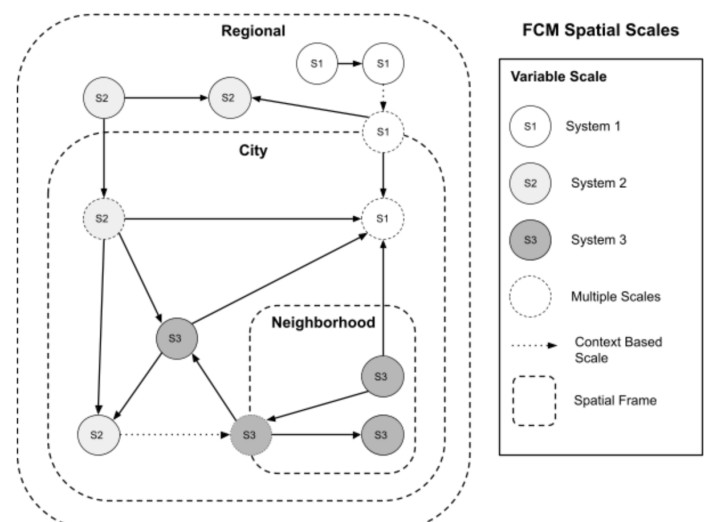

**Figure 3.** Fuzzy cognitive maps (FCM) have two distinct types of spatial scale: a spatial frame and system dynamic scales. The spatial frame represents the spatial scale in which the FCM was framed. The system dynamic scales are the contextual spatial scales of the variables and relationships that make up the FCM. Each variable has a spatial scale that depends on its relationships with other variables in the system.

By coupling the spatially explicit survey results with the regional FCM (RFCM), we generated spatially filtered FCMs (SFFCM) for each stormwater problem cluster (Figure 4). We first parsed the text of the responses to the following survey questions for each cluster:

- Why might this location be a problem?
- What is the greatest barrier to resolving this problem?
- What actions can be taken to "fix" this problem?

The parsed text was then matched to variables found in the regional FCM. Additionally, each word was cross-referenced to words used before the FCM variables were coded to account for a broader range of language. The resulting variables were then connected based on the relationships defined in the regional FCM. The resulting FCM was a spatially filtered FCM that depicts a downscaled representation of the regional FCM. However, the spatially filtered FCM represents only a portion of the regional FCM and does not depict the FCM of a stormwater problem cluster. To represent the FCM of the stormwater cluster, a manually coded FCM (MFCM) was generated by the research team from the same

stakeholder survey response data used to create the spatially filtered FCM [3]. To develop a MFCM, we first identified the variables from the text survey responses. If a response matched a variable used in the regional FCM, we used the same term to allow for a later comparison of variables. For variables not found in the regional FCM, we introduced a new variable. Relationships between variables were inferred based on the context of the survey text, and we set all weights to either $-1$.

The spatial frame and system dynamic scale were assigned to the spatially filtered and manually coded FCM based on how the survey responses indicated the spatial scale of the problem and impact. The spatial frame is the problem scale because that is how the survey framed the question. The system dynamic scales are approximated by mapping the problem and impact scales to the FCMs' variables based on their relationships. Receiver variables represent the cluster's problem (e.g., water quality, flooding, etc.) and have the spatial scale of the impact of the problem. Transmitter variables represent the barriers (e.g., upstream development, impervious surfaces, etc.) to the solutions and have the spatial scale of the problem. The survey solutions are often characterized by central variables with a high out-degree, but are not transmitters. The spatial scale of the solutions depends on what the solution is addressing. If the solution addresses the problem directly (e.g., flooding, etc.), the spatial scale is the impact scale, and if the solution addresses the barrier of the problem (e.g., urban development, etc.), the spatial scale is the problem scale.

After the spatially filtered and manually coded FCMs were created for a cluster, we compared them to each other by examining the models' structure and variable similarity and by running FCM scenarios identified by stakeholders in the survey.

To begin, we calculated the Jaccard similarity coefficient between (1) the regional and spatially filtered FCMs, (2) the regional and manually coded FCMs, and (3) the spatially filtered and manually coded FCMs to compare how similar the variables of each FCM were to one another [26]. When comparing (1) the regional and spatially filtered FCMs, the similarity coefficient represents the percentage of the regional FCM captured in the spatially filtered FCM. When the spatially filtered FCM contains a high proportion of the regional FCM's variables, it indicates that the stakeholders may consider more causal relationships that cross spatial scales. In contrast, if a spatially filtered FCM contains a low percentage of variables from the regional FCM, the stakeholders may consider fewer causal relationships at a smaller spatial extent. The similarity index between (2) the regional FCM and the manually coded FCM indicates how well the regional FCM represents an actual cluster's problem as described by stakeholders. When the FCMs have low similarity, it suggests that the regional FCM does a poor job of representing the problem at the cluster's spatial scale. In contrast, a high level of similarity suggests the regional FCM performs well at modeling the problem at the cluster's spatial scale. Finally, when (3) the spatially filtered and manually coded FCMs are compared, the similarity coefficient represents how well the spatially filtered FCM captures the actual problem modeled in the manually coded FCM. If the similarity coefficient is low, it indicates that the spatially filtered FCM does of poor job of modeling the cluster's problem. If the similarity coefficient is high, the spatially filtered FCM does a good job of modeling the cluster's problem.

Next, we ran "what if" scenarios on the spatially filtered, manually coded, and regional FCM of each cluster to examine how the modeled outcomes varied when run with FCMs created at different spatial scales. To begin, we identified scenarios defined by stakeholders in the spatially explicit survey when asked what solutions could be taken to address the stormwater problem. For example, below is a survey response suggesting a solution to reduce flooding:

> *"I think that the tributaries feeding into the [C]rabtree Creek need to be controlled by increasing riparian vegetation and decreasing the paved surface."*

We then reframed the solution as a "what if" question that fit the framework of the FCMs:

> *What if we increase riparian buffers and decrease impervious surfaces?*

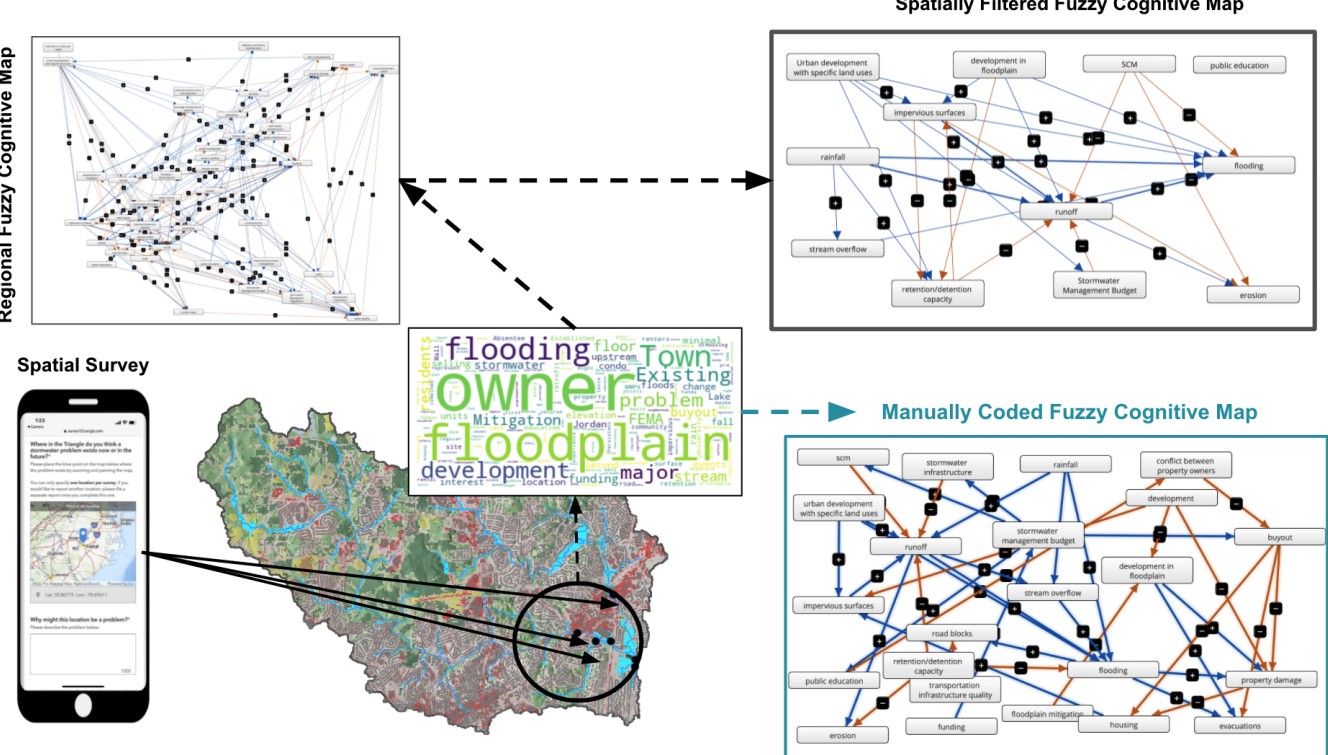

**Figure 4.** By coupling spatially explicit survey data with a regional fuzzy cognitive map of stormwater management, we generated spatially filtered FCMs to explore the role of spatial scale in FCMs. The process starts by creating a regional FCM and collecting spatially explicit survey data about the same system. The text responses from the survey are then parsed and matched to variables used in the regional FCM. The result from this process is a spatially filtered version of the regional FCM. A manually coded FCM is also independently generated from the survey responses to capture variables and relationships not identified in the regional FCM.

We ran each scenario by activating the variables by either −1 (a little) when decreasing or 1 (a lot) when increasing a variable's activation state. We chose only to activate variables using −1 and 1 to reduce complexity when comparing models. We then compared the scenario results for each FCM by identifying if the desired effect of the proposed solution was modeled as expected across all of the models. We also examined other impacts the solution had across the FCMs to identify gaps in stakeholder causal reasoning across spatial scales.

## 3. Results

### 3.1. Individual Fuzzy Cognitive Maps of Stormwater Management

During Workshop 2, 21 individual FCMs were created and returned for analysis. The 21 analyzed FCMs contained a total of 156 variables before basic corrections of spelling and tense. After grammatical errors were removed, two members of the research team used inductive coding to combine similar terms, reducing the total number of unique components by 21% to 124. The accumulation curve of the total number of unique components shows that the number of new variables added by each map did not reach a stable state (trend line did not flatten out; (Figure 5, left side). The accumulation curve of the number of unique components added per new FCM shows that the number of new variables captured with each new map reached a threshold near 5 (Figure 5, right side).

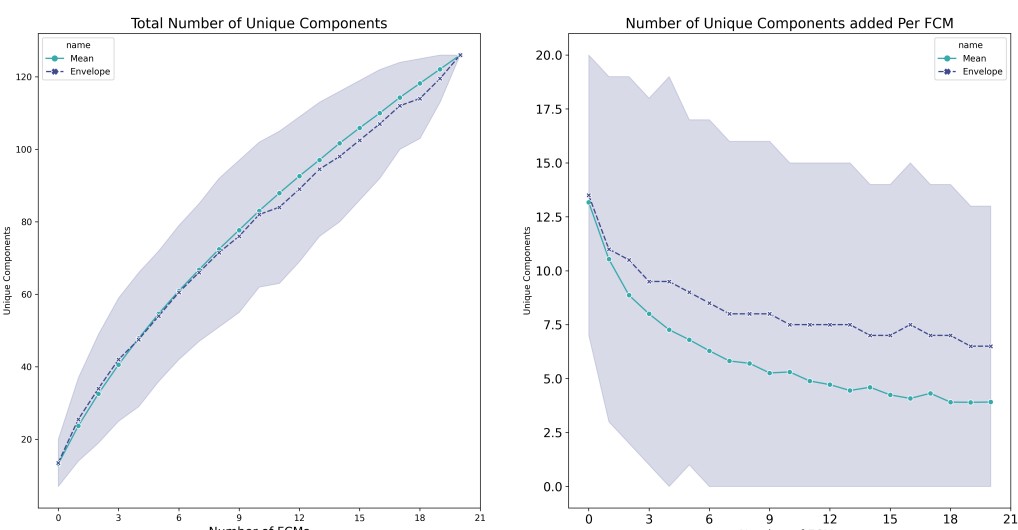

**Figure 5.** Fuzzy cognitive map (FCM) component accumulation curves representing the total number of unique components for all stakeholder FCMs (**left**) and the number of new unique components added with each additional FCM (**right**). The total number of unique components curve indicates the system has not yet reached a stable state. The number of unique components added per FCM curve indicates that a threshold of about five new variables is added with each additional FCM.

### 3.1.1. Analysis of Academic and Government FCM Indices

The total number of variables found in each FCM was normally distributed, with the mean stakeholder FCM containing 13.2 ± 3.5 variables (N) and 20.1 ± 8.0 connections (C) (Table 1). The individual FCMs contained more transmitter variables than receivers on average. However, the individual transmitter and receiver variables both contained a high level of variability (Table 1). Stakeholders' individual FCMs had a mean hierarchy index of 0.8 ± 0.3, indicating that the majority of stakeholders' FCMs were more hierarchical than democratic, which could indicate that the stakeholders are less likely to perceive the system as easily adaptable [6,28,50].

Statistical analysis of the academic and government stakeholder groups' FCM indices indicates no statistically significant differences ($p \leq 0.05$) in the model structures (Table 1). Stakeholders from industry and NGOs were not compared with the other stakeholder groups due to their sample size.

Among the 21 stakeholder-defined FCMs, 124 unique variables were mentioned, with 43 (35%) mentioned in more than one FCM after coding. The most frequently mentioned variables were flooding and impervious surfaces, which were found in 76% of the stakeholder FCMs. Stakeholders included 47 unique transmitter variables; the most frequent transmitter was rainfall, which appeared in 48% of the FMCs. The stakeholders also included 31 unique receiver variables; water quality and flooding were the most common, appearing in 24% of FCMs. Stormwater runoff and flooding were the most common central variables, with runoff appearing in 24% of the FCMs and flooding in 19%. The most frequent variables were more likely to be transmitter or receiver variables, indicating that the stakeholders agreed on the driving forces and outcomes of the stormwater management system but lacked agreement on ordinary variables.

**Table 1.** The 21 individual stakeholder fuzzy cognitive map structures were compared using graph indices. The FCMs were filtered into academic and government stakeholder groups for comparison. Stakeholder groups' FCM indices were compared using standard statistical tests such as ANOVA or Mann–Whitney. No statistically significant differences ($p \leq 0.05$) were found after comparing FCM indices from academic and government stakeholder groups *(value ± SD)*.

| | | | | | |
|---|---|---|---|---|---|
| **Fuzzy Cognitive Maps Indices** | | | | | |
| | **All** | **Academic** | **Government** | **ANOVA** **(*p*-Value)** | **Mann-Whitney** **(*p*-Value)** |
| Maps | 21 | 7 | 10 | | |
| Variables (N) | 13.2 ± 3.5 | 12.7 ± 4.3 | 13.5 ± 3.6 | 0.6909 | - |
| Connections (C) | 20.1 ± 8.0 | 16.9 ± 5.5 | 22.6 ± 10.0 | 0.1924 | - |
| Ordinary | 7.5 ± 2.6 | 8.1 ± 2.5 | 6.8 ± 2.5 | 0.2947 | - |
| Transmitters (T) | 3.6 ± 2.6 | 2.6 ± 2.2 | 4.4 ± 3.2 | - | 0.0667 |
| Receivers (R) | 2.1 ± 1.7 | 2.0 ± 2.4 | 2.3 ± 1.7 | - | 0.3440 |
| Complexity (R/T) | 0.8 ± 1.1 | 1.1 ± 1.7 | 0.7 ± 0.9 | - | 0.3831 |
| Hierarchy | 0.8 ± 0.3 | 0.7 ± 0.3 | 0.8 ± 0.2 | - | 0.1845 |
| Density | 0.1 ± 0.1 | 0.1 ± 0.1 | 0.2 ± 0.1 | - | 0.5 |

While both academic and government stakeholder groups used a wide variety of language to define stormwater management, academic stakeholders had less agreement in the variables used to describe the system. Variables used more than once comprised only 40% of the total variables used by academics and 63% of the variables used by government personnel. The Pearson's Chi-Square Test was used to compare variables mentioned more than once by each group. The test identified a significant relationship ($\alpha < 0.05$) between stakeholder groups and variables ($p = 0.019$). Specific variables of note included urban development with specified land use, which appeared in 70% of the ten government FCMs and none of the academic FCMs, and climate change, which was mentioned in 57% of academic FCMs and 20% of government FCMs (Table A1).

3.1.2. Regional Fuzzy Cognitive Map

By aggregating together all stakeholder FMCs, a social cognitive model was created containing 124 variables, 347 connections, 18 receivers, and 27 transmitters (Table 2). We then simplified this model to develop the regional FCM by removing variables that were mentioned only once by stakeholders before aggregating the individual FCMs, creating a conservative but more interpretable model (Figure 6) with 43 variables and 172 connections (Table 2). The regional FCM had a hierarchy of 0.08, indicating that the aggregated model may provide a more democratic model of stormwater management that is more adaptable to environmental changes than an individual FCM [6,28,50] (Table 2).

**Table 2.** The social cognitive model and regional FCM were generated by aggregating the individual stakeholder FCMs. The regional FCM used only variables that were mentioned more than once by stakeholders and is therefore a conservative representation of the stakeholders' perception of stormwater management in the Research Triangle, North Carolina, at the time of the workshop.

| | **Social Cognitive Model** | **Regional FCM** |
|---|---|---|
| Maps | 21 | 21 |
| Variables (N) | 124 | 43 |
| Connections (C) | 347 | 172 |
| Ordinary | 79 | 39 |
| Transmitters (T) | 27 | 3 |
| Receivers (R) | 18 | 1 |
| Complexity (R/T) | 0.7 | 0.33 |
| Hierarchy | 0.2 | 0.08 |
| Density | 0.02 | 0.09 |

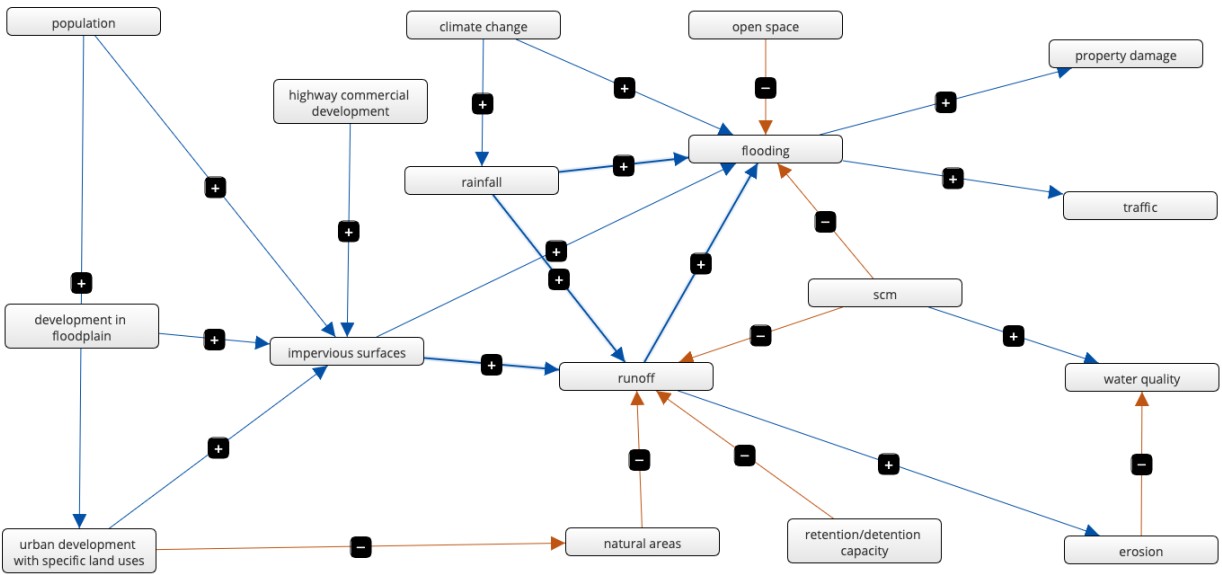

**Figure 6.** The regional FCM shown is filtered to exclude weak causal connections and reveal the core model structure. The model shows that stakeholders perceive population and climate change as two driving forces contributing to increased impervious surface and rainfall. Stakeholders also included causal relationships for management strategies, such as decreasing development in the floodplain to reduce impervious surfaces, adding additional retention/detention capacity, and other stormwater control measures to reduce runoff and flooding.

The regional FCM's central features show that flooding, water quality, and runoff are the most influenced variables in the system, while impervious surfaces, urban development with specific land uses, and rainfall are the most influential variables (Figure 7). For example, changes to urban development or impervious surfaces (high out-degree) during a scenario simulation will significantly affect the system compared to changes in stream/ecosystem health (high in-degree).

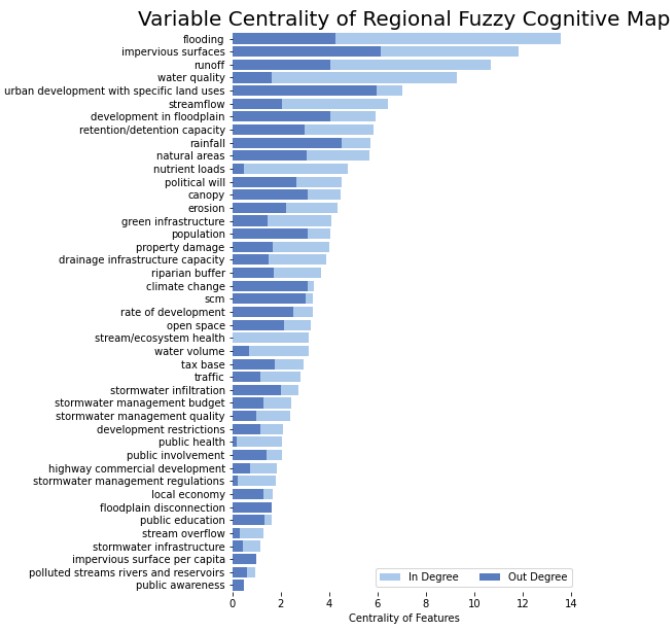

**Figure 7.** Flooding is the most central feature found in the regional FCM. Impervious surface and urban development influence other variables the most, as evident by their out-degree, and flooding and water quality are the most influenced variables.

### 3.2. Understanding Spatially Explicit Stormwater Problems

During Workshop 2, stakeholders located 31 stormwater problem areas in the Triangle, NC. At each location, they identified why the location is a problem, what barriers are preventing the problem from being solved, and proposed solutions to fix the problem. Stakeholders indicated flooding as the most common problem and existing urban development as the most common barrier to resolving the issue. The strategies most frequently identified by stakeholders to "fix" the identified stormwater problems included improvements to infrastructure and upstream development restrictions. The stakeholders most frequently considered the problems somewhat serious (67.74%) and very hard to solve (45.61%) (Figure 8). Stakeholders reported that identified problems currently exist and will likely exist in the future (80.65%). However, no stakeholders located an area where they predicted a future stormwater problem occurring. The problems most frequently occur at annual (61.29%) and monthly (32.26%) time intervals, while the impact of these problems last from a day to a week (Figure 8). The most common spatial scale of the drivers of the problems is the neighborhood (45.16%), closely followed by city-wide (41.94%). The most common spatial scale of the impact of the problems is the neighborhood (54.84%) (Figure 8). By combining the spatial and temporal scales of the stormwater problems, we identified the most common spatial-temporal scale as city-wide to neighborhood-level drivers (e.g., urban development) with neighborhood-level impacts (e.g., flooding) that occur annually and last a day.

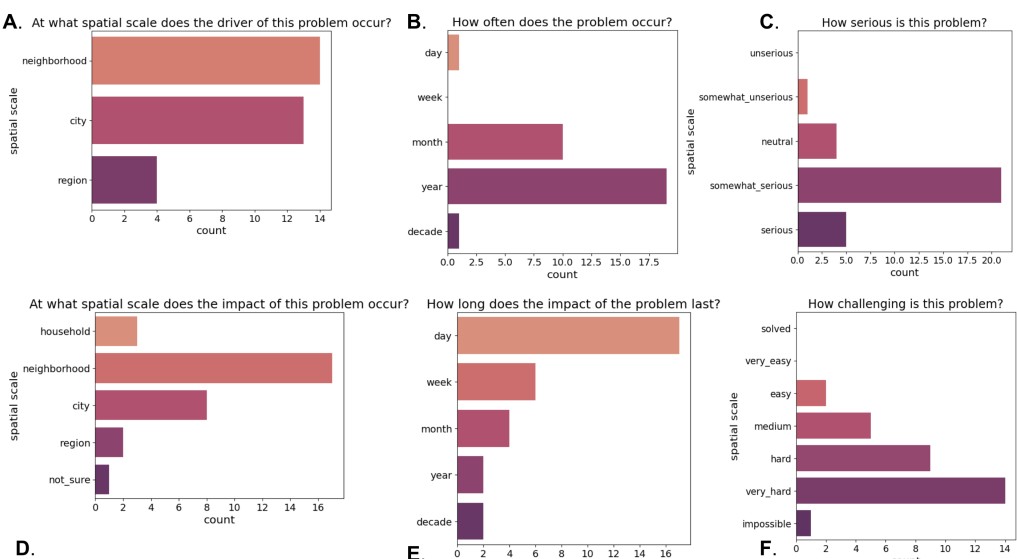

**Figure 8.** Stakeholders identified 31 stormwater problem areas addressing issues surrounding the spatial-temporal scale of the problems occurring (**A**,**B**) and the socio-environmental impact of the problem (**D**,**E**). Additionally, stakeholders identified their perceptions of the seriousness of the problem (**C**) and how challenging the problem is to solve (**F**).

### 3.3. Problem Hotspots: Spatial Characteristics and Solutions

Three clusters representing different problem areas were identified within Orange, Durham, and Wake Counties. The clusters consisted of 45% of the locations identified by stakeholders. Cluster 1 (University Mall) is located in the Cape Fear River Basin in the Town of Chapel Hill, and Clusters 2 (Crabtree Valley Mall) and 3 (North Carolina State University) are located in the Neuse River Basin in the City of Raleigh (Figure 9). Annual flooding was the primary problem identified at all of the clusters, with existing upstream development commonly listed as the most significant barrier to solving the problem. The impacts of the flooding ranged from household-level to city-wide and lasted from a day up to a year. Each of the problems currently exists, and stakeholders believe the problems

will continue to exist into the future. All of the clusters are considered somewhat serious problems that are hard to solve.

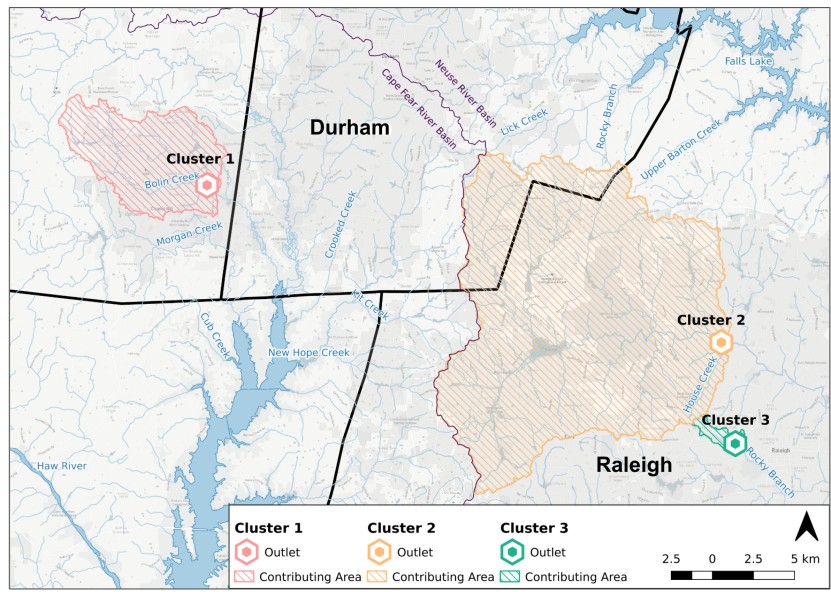

**Figure 9.** Three clusters representing stormwater problems in the Research Triangle, North Carolina, were identified from the spatial survey stakeholders completed during Workshop 2.

For each cluster's upstream contributing area, we evaluated stakeholder descriptions of the problems, barriers, and solutions using the spatial characteristics of land cover, development patterns, and simulations of pluvial and fluvial flooding. We also examined the stakeholders' assignments of the spatial scale of the drivers and impacts for each cluster. We then tested stakeholder proposed solutions to the clusters' problems using the regional FCM, spatially filtered FCM, and manually coded FCM for each cluster.

### 3.3.1. Cluster 1—University Mall

Cluster 1 consists of five survey points located near University Mall in Chapel Hill, NC. The cluster is part of the B. Everett Jordan Lake-New Hope River watershed near the intersection of Bolin Creek and Booker Creek where they converge into Little Creek. The cluster is in a water supply watershed for Jordan Lake and is in the jurisdiction of the Town of Chapel Hill, NC. The upstream contributing area is 48.1 km$^2$ and is 56% developed (NLCD 2016) with major upstream development occurring post-1980 (Figure 10).

Problem

Stakeholders described Cluster 1 as prone to annual flooding with effects felt at a neighborhood scale from a day to a month. The flooding is caused by a large amount of upstream development that accumulates at the intersection of Bolin and Booker Creeks. The upstream contributing area is part of both the Town of Chapel Hill and the Town of Carrboro, NC, and contains 356 buildings within the floodplain. These buildings include Camelot Village and University Mall, which stakeholders specifically identified as locations that frequently flood. University Mall was opened in 1973 before an additional 14 km$^2$ of land was developed in the upstream contributing area, increasing the total developed area by 110% (Figure 10). We validated stakeholder concerns about flooding by running a simulated one-hour, 100-year precipitation event for the upstream contributing area. The simulation indicated runoff accumulating to depths of 2.6 m and showed pluvial flooding occurring both within and outside the floodplain (Figure 11). A one-meter maximum inundation was also simulated and closely matched the FEMA floodplain map and stakeholder descriptions of the impact of the problem. In both simulations, Camelot

Village and University Mall were flooded, indicating that the spatial characteristics of the problem align with stakeholder concerns.

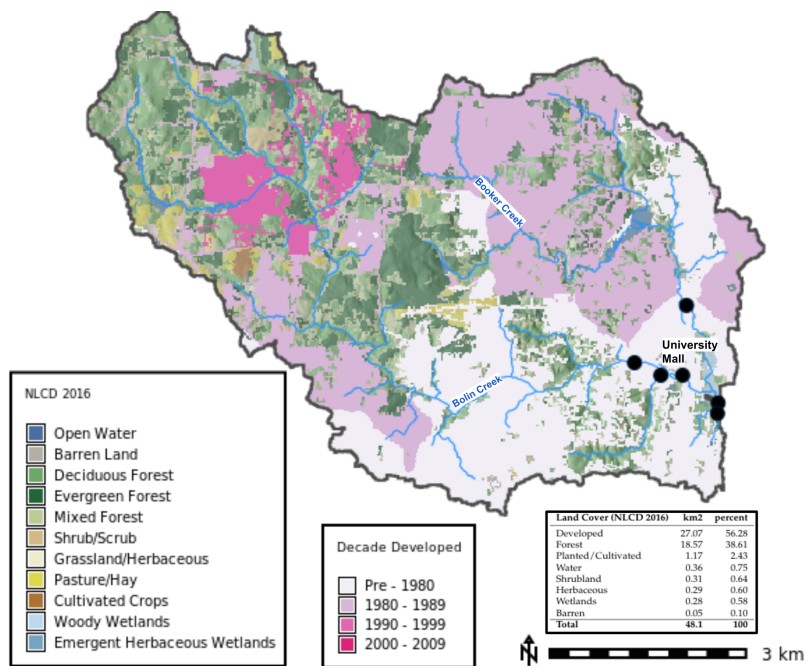

**Figure 10.** Cluster 1—Cluster 1 (University Mall) has an upstream contributing area of 48.1 km$^2$ that is 56% developed. The cluster was developed before the 1980s, with University Mall opening in 1973. After this initial development, an additional 14 km$^2$ of developed land was added upstream, contributing to the current flood conditions identified by stakeholders.

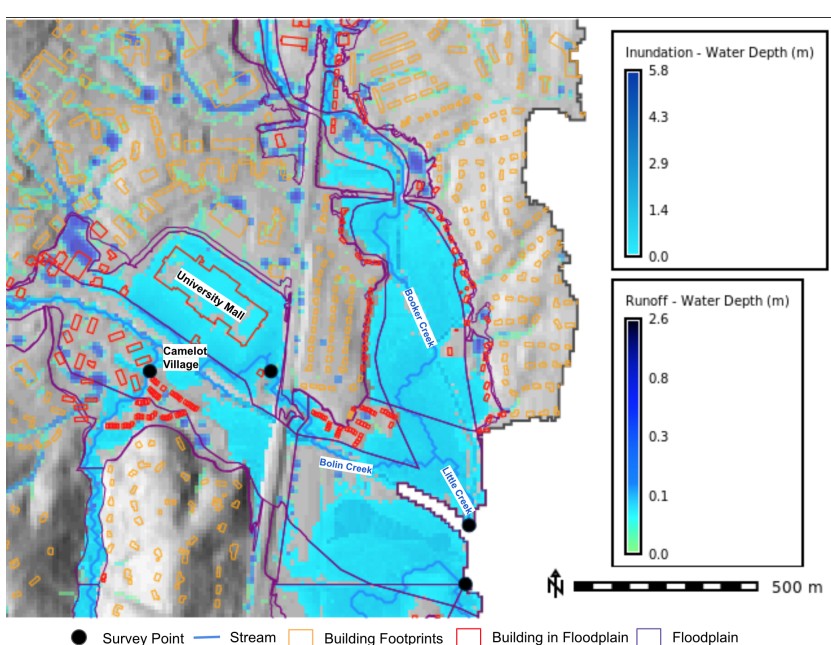

**Figure 11.** Cluster 1—University Mall and Camelot Village are shown inside the floodplain and flooded during a simulated 100-year precipitation event. Surface water runoff accumulated to depths up to 2.6 m, and the one-meter inundation extent shows that the locations are prone to flooding.

Barrier

Stakeholders identified existing development near and within the floodplain, the cost of buying out properties in the floodplain, the cost of improving infrastructure, and

disagreement among property owners regarding property acquisition and removal (i.e., floodplain buyouts [51]) as barriers preventing the problem from being resolved. When examining the spatial characteristics of the barriers, we identified existing development in the floodplain by comparing FEMA flood maps with land cover. The floodway was 40% developed, the 100-year floodplain was 40% developed, and the 500-year floodplain was 63% developed. However, the majority of the developed floodplain is open space and low-intensity development. Other barriers like the costs of new infrastructure, floodplain buyouts, and disagreement between property owners are not easily quantified by the spatial characteristics of the upstream contributing area.

Solution

Solutions for this problem area included increasing retention/detention capacity, development restrictions, and public education about the risk of flooding. These solutions highlight two overall strategies: managing upstream runoff that is causing the flooding or moving buildings and people to limit the impacts of flooding via Town- or FEMA-funded floodplain buyouts. These strategies were translated into the following "what if" scenarios in order to be tested using the FCM:

1. Increase retention/detention capacity
2. Increase public education
3. Increase stormwater control measures (SCM)
4. Increase stormwater management budget
5. Increase development restrictions

To run the scenarios, a spatially filtered and manually coded FCM was created for the cluster. The models represent systems with drivers at a city-wide spatial scale and impacts at the neighborhood level. The spatially filtered FCM contained 13 variables with 30 connections representing 30% of the regional FCM. It represented two problems (flooding and erosion) as receiver variables and four barriers (urban development with specific land uses, development in the floodplain, SCM, and rainfall) as transmitter variables. The manually coded FCM contained 23 variables connected by 38 causal relationships and represented 35% of the regional FCM with a similarity coefficient of 0.29. The manually coded FCM had eight unique variables not represented in the regional FCM. The manually coded and spatially filtered FCMs had a similarity coefficient of 0.56, and all of the spatially filtered FCM variables were found in the manually coded FCM.

We ran two scenarios on the regional, spatially filtered, and manually coded FCMs to compare stakeholders' perceived solutions to flooding issues in Cluster 1 at various spatial scales. In the first scenario, we increased retention/detention capacity, and in the second scenario we increased public education. When we increased retention/detention capacity, stormwater runoff and erosion decreased across all three models. However, flooding was reduced only in the spatially filtered FCM and manually coded FCM, indicating that the regional FCM contained causal relationships that were not present when the model was spatially filtered. Additionally, the model output for the regional FCM showed positive environmental effects, such as an increase in water quality and a decrease in nutrient loading that were not present in the spatially filtered or manually coded FCMs. Furthermore, the manually coded FCM indicated reductions in evacuations, property damage, and roadblocks. These results suggest that the regional FCM misses causal relationships surrounding public safety. In scenario two, public education was increased for each of the FCMs. The results from the manually coded FCM showed a decrease in conflict between property owners and an increased likelihood of buyout. However, public education in the spatially filtered FCM contains no connections to other variables, indicating that public education addresses another issue or has another meaning in the regional FCM. When public education was increased in the regional FCM, political will, stormwater management regulations, and stormwater management quality increased, indicating public education at a regional scale has different meanings or intent.

### 3.3.2. Cluster 2—Crabtree Valley Mall

Cluster 2 is located at Crabtree Valley Mall in the floodplain of Crabtree Creek in the Crabtree Creek watershed in the City of Raleigh, North Carolina. The problem was identified by six stakeholders and described as having drivers such as existing development at the neighborhood scale and impacts like flooding and traffic at the neighborhood to city-wide scale.

Problem

Stakeholders indicated that the major problem for this cluster is flooding that causes property damage and flooding over the roadway that causes traffic. Flooding occurs annually, with impacts of the event lasting from a day up to a week in the surrounding neighborhood and city. Stakeholders' concerns surrounding flooding were validated by running models to simulate a 100-year precipitation event to examine pluvial flooding and a one-meter maximum inundation event (Figure 12). These models indicated that stormwater runoff could cause pluvial flooding up to 3 m and fluvial flooding up to 22.1 m. The models also indicated flooding over the roadway, but the effects of increased traffic were not examined via geospatial simulation for this study.

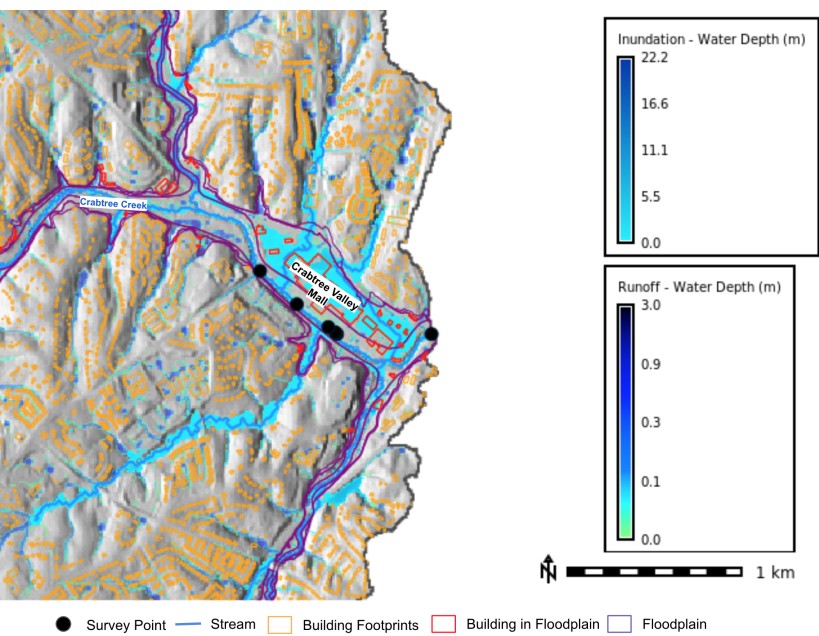

**Figure 12.** Cluster 2—During a simulated 100-year precipitation event, overland flow accumulated to depths up to 3m. One-meter maximum inundation extent shows that the location is prone to flooding.

Barrier

The barriers identified by stakeholders as preventing this problem from being addressed were a combination of the mall having a large upstream contributing area, existing development in the floodplain, the cost of floodplain mitigation and new infrastructure, and the prioritization of the problem. Spatial analysis of the cluster supports stakeholders' claims about the mall's large upstream contributing area and existing development in the floodplain. Cluster 2 has a 252.5 km$^2$ upstream contributing area that is 61% developed. The problem hotspot (Crabtree Valley Mall) was developed inside the floodplain in the 1970s before an additional 122.83 km$^2$ of developed land was built upstream (Figure 13). This new development represents a 384% increase in developed land over 40 years. The upstream contributing area has 19.17 km$^2$ within the floodplain, 30% of which is developed. The stakeholders characterized the driver scale of the cluster at the neighborhood level, which could indicate they are underestimating how much development is draining into the problem area.

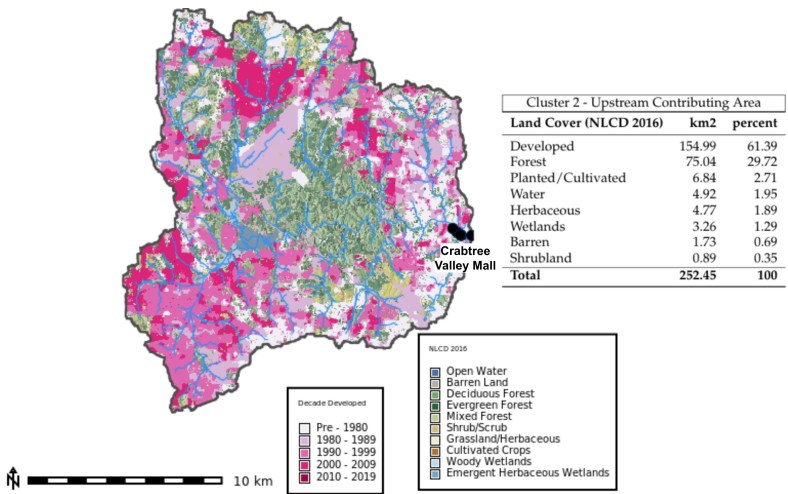

| Cluster 2 - Upstream Contributing Area | | |
|---|---|---|
| **Land Cover (NLCD 2016)** | **km2** | **percent** |
| Developed | 154.99 | 61.39 |
| Forest | 75.04 | 29.72 |
| Planted/Cultivated | 6.84 | 2.71 |
| Water | 4.92 | 1.95 |
| Herbaceous | 4.77 | 1.89 |
| Wetlands | 3.26 | 1.29 |
| Barren | 1.73 | 0.69 |
| Shrubland | 0.89 | 0.35 |
| **Total** | **252.45** | **100** |

**Figure 13.** Cluster 2—Cluster 2 has a 252.5 km$^2$ upstream contributing area that is 61% developed. The problem hotspot (Crabtree Valley Mall) was developed inside the floodplain in the 1970s before an additional 122.83 km$^2$ of developed land was built upstream. This new development represents a 384% increase in developed land over 40 years.

Solution

Stakeholders proposed a variety of solutions to the problem, including increasing development restrictions and improving stormwater and transportation infrastructure. The desired effects of these scenarios are to decrease flooding, specifically over the roadway to reduce traffic. By matching variables from the stakeholder responses to the regional FCM, we developed a spatially filtered FCM and manually coded FCM to test the stakeholder solutions with the following scenarios:

1. Increase development restrictions
2. Decrease impervious surfaces
3. Increase floodplain mitigation
4. Increase riparian buffer
5. Increase stormwater infrastructure
6. Increase transportation infrastructure

The regional FCM was filtered to a spatially filtered FCM with neighborhood-scale transmitters such as urban development and neighborhood- to city-wide-scale receivers like traffic. The spatially filtered FCM consisted of 11 variables and 21 connections representing 26% of the regional FCM with a similarity coefficient of 0.26.

The manually coded FCM included 26 variables and 49 connections representing 28% of the variables found in the regional FCM. Barriers such as the upstream contributing area, urban development, and stormwater management budget appeared in the system as transmitter variables. The problems such as flooding and traffic were modeled as ordinary variables instead of receivers due to the number of causal relationships. Fourteen variables from the manually coded FCM were not identified in the regional FCM or the spatially filtered FCM, giving it a similarity coefficient of 0.21 with the regional FCM and a similarity coefficient of 0.37 with the spatially filtered FCM.

For Cluster 2, we ran two stakeholder-defined scenarios to compare differences in stakeholders' perceived solutions to flooding at various spatial scales using the spatially filtered FCM, manually coded FCM, and regional FCM. The first scenario increased development restrictions, and the second scenario increased stormwater infrastructure. The first scenario shows that increasing development restrictions decreases development in the floodplain, runoff, and flooding across all three FCMs. However, the regional FCM shows environmental impacts not identified by the spatially filtered FCM or manually coded FCM, such as an increase in water quality, stream health and natural areas, and decreases in nutrient loading. These results indicate that stakeholders' causal reasoning

of how development restrictions affect flooding remains constant across neighborhood and regional spatial scales. However, the secondary effects of development restrictions such as environmental impacts on water quality and stream health are overlooked when stakeholders focus on stormwater problems at a neighborhood scale.

In the second scenario, we increased stormwater infrastructure, and all three models agreed that stormwater infrastructure decreases flooding. However, the manually coded FCM identified a decrease in flooding over the roadway and traffic not captured by the spatially filtered FCM or regional FCM. The manually coded FCM also included more specific stormwater management goals, such as increased control tributaries and increased flow capacity under bridges. These results indicate that spatial scale impacts stakeholders' causal reasoning of the influences of stormwater infrastructure on flooding by changing the level of detail (spatial grain) at which the stakeholders consider the problem. In addition, stakeholders' personal and professional experience with the stormwater infrastructure of a location influences their level of detail.

### 3.3.3. Cluster 3—North Carolina State University

Cluster 3 consists of three stakeholder responses on North Carolina State University's (NCSU) main campus. The cluster is located in Wake County and is in the Walnut Creek–Neuse River watershed near the Rocky Branch Creek.

### Problem

Stakeholders described the problem as household-level flooding caused by excess drainage into insufficient or damaged stormwater infrastructure. For example, one stakeholder stated the problem as

*"50 acres of land draining to a single 30″ pipe"*

The driver of the problem is at the neighborhood-level, indicated by its upstream contributing area of 2.99 km$^2$. The upstream contributing area is highly developed (93%), containing 896 buildings and a dense underground utility network (Figure 14).

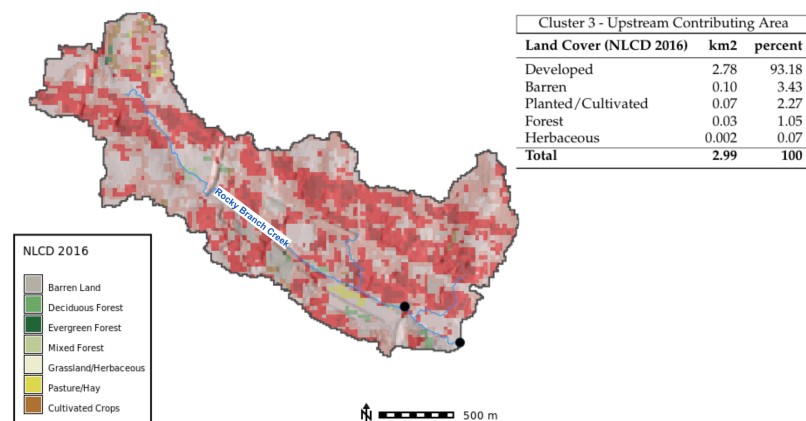

| Cluster 3 - Upstream Contributing Area | | |
|---|---|---|
| **Land Cover (NLCD 2016)** | **km2** | **percent** |
| Developed | 2.78 | 93.18 |
| Barren | 0.10 | 3.43 |
| Planted/Cultivated | 0.07 | 2.27 |
| Forest | 0.03 | 1.05 |
| Herbaceous | 0.002 | 0.07 |
| **Total** | **2.99** | **100** |

NLCD 2016
- Barren Land
- Deciduous Forest
- Evergreen Forest
- Mixed Forest
- Grassland/Herbaceous
- Pasture/Hay
- Cultivated Crops

500 m

**Figure 14.** Cluster 3—Cluster 3 is located at North Carolina State University in Raleigh, North Carolina and has an upstream contributing area of 2.99 km$^2$ that is 93% developed.

Pluvial flooding was modeled using a 100-year precipitation event and indicated that water can accumulate up to depths of 1.15 m. A one-meter maximum inundation was also simulated, showing 2000 m$^2$ maximum flood extent with water depths reaching 4.6 m (Figure 15).

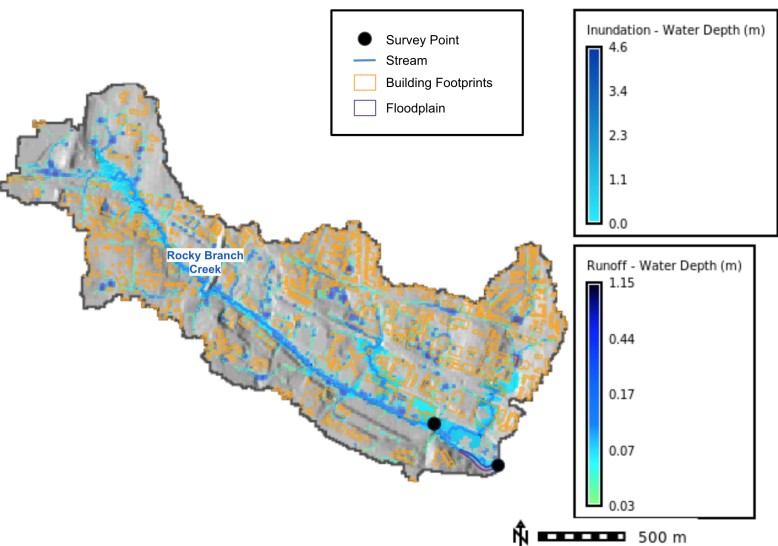

**Figure 15.** Cluster 3—Maximum inundation and 100-year precipitation events were simulated for Cluster 3, indicating pluvial flooding at depth up to 1.15 m and fluvial flooding up to 4.6 m deep.

Barrier

Barriers included specific stormwater infrastructure repairs being a low priority with the City of Raleigh (e.g., collapsed storm drain). Additionally, fixing the problem was described as a complex engineering problem because of existing development and a dense underground utility network.

*"How to fit a suitable sized pipe over a 10 foot tall underground tunnel"*

Solution

Stakeholders' proposed solutions to this problem included improving and repairing the stormwater infrastructure. We tested the stakeholders' solutions by running a scenario that increased stormwater infrastructure with the spatially filtered, manually coded, and regional FCM. The spatially filtered FCM for Cluster 3 was small, capturing a generalized model of the problem (flooding) from rain events due to insufficient stormwater infrastructure. The FCM contained three variables representing 7% of the regional FCM. The manually coded FCM contained 11 variables and 17 causal connections focusing on specific engineering and stormwater management challenges. All three of FCMs for Cluster 3 shared the variables stormwater infrastructure, rainfall, and flooding but had very low similarity coefficients. The regional FCM had a similarity coefficient of 0.06 with the manually coded FCM and 0.07 with the spatially filtered FCM.

The regional FCM showed decreases in flooding, nutrient loading, and runoff, and an increase in retention/detention capacity. However, the spatially filtered FCM results showed that relying on the regional model alone provided a coarse view of the problem and solution and indicated only that flooding will decrease. In comparison, the manually coded FCM considered the problem at a much finer spatial scale and considered the priority and complexity of specific stormwater engineering challenges.

## 4. Discussion

Disparate urban areas face challenging stormwater management problems given rapid urbanization and increasingly severe weather events. As the complexity and severity of socio-environmental challenges grows, finding shared understanding between diverse stakeholder groups becomes ever more critical to address "wicked" problems. Cognitive models paired with spatial data offer a way to build that understanding, revealing the wide variety of language used to describe problems and the complexities of how spatial scale influences decision-making [3,4].

### 4.1. A Shared Understanding of Stormwater Management

In this study, stakeholders used a wide range of language (124 variables) to model stormwater management. The FCM variable accumulation curves indicated that stakeholders do not possess a strong shared vocabulary when discussing stormwater management and that the system was not entirely represented by the stakeholders involved. Additionally, statistically significant differences appeared during variable selection between academic and government stakeholders. For example, the variable urban development with specific land uses was not mentioned in any academic FCMs but appeared in 70% of the government stakeholders' FCMs. In contrast, 57% of the academic and 80% of the government FCMs mentioned the variable impervious surfaces. These results indicate that focusing on regulating impervious surfaces may provide a more successful path to managing surface water runoff than focusing on urban development.

### 4.2. The Role of Spatial Scale on Fuzzy Cognitive Maps

Spatial scale changes the perceptions and patterns of environmental and ecological processes [52]. However, explicitly setting spatial scale is often overlooked when modeling SES [52–54]. This is especially true when developing FCMs during participatory modeling [3]. Participants are typically asked to consider the components of a system but not necessarily at a specific location or spatial-temporal scale [55]. Gray et al. (2014) asked stakeholders to draw the spatial extent of the FCM they developed as a polygon. However, they did not examine if the spatial extent (i.e., spatial framing) impacted the FCM itself or the spatial scale of the variables and connections. Using a polygon instead of point data would be another valuable source of comparison to stakeholders' perceptions of scale in stormwater management and the underlying hydrological properties of the area. Other options include stakeholders' "spray painting" areas on a map to provide 'fuzzy' spatial data [56]. The Spraycan method has proven an effective means to capture uncertainty sometimes found in stakeholder-provided point data. Our study used point data because it was the only available option provided by Survey123 for ArcGIS. However, by connecting the point data to the local hydrology, we could compare the spatial extent of the upstream contributing area to the survey responses provided by stakeholders. Given the importance of scale for modeling problems and solutions with geospatial data, we examined the effect of spatial scale on FCMs and demonstrated how spatial scale impacts the variables and relationships described in FCMs addressing stormwater management in the Triangle region of North Carolina.

The regional FCM illustrated that stakeholders perceive water quality, property damage from flooding, and traffic as the core problems directly influenced by stormwater management. In contrast, stakeholders perceived population growth and climate change as the significant drivers of stormwater management in the Triangle. The results indicated that increasing stormwater control measures are essential for reducing runoff and flooding to minimize property damage, protect water quality, and prevent increased traffic at a regional scale. However, 'stormwater control measure' is a vague term that represents a variety of different management strategies [57], indicating that regional-scale FCMs are too coarse to understand specific stormwater management strategies. Stakeholders most often located spatially explicit stormwater problems at neighborhood and city spatial scales instead of regional. As a result, stakeholders' perceptions of stormwater management were filtered based on the scale of the problem. For example, environmental issues surrounding climate change, water quality, and stream/ecosystem health were not mentioned by stakeholders when identifying spatially explicit stormwater problems, despite identification of problem locations upstream of drinking water supplies. However, these issues were among the most common receiver variables in the regional FCM. These results could indicate that environmental issues such as climate change and water quality are often overlooked when addressing spatially explicit stormwater problems at spatial scales smaller than a regional scale. Similarly, issues surrounding public safety and interpersonal conflicts were not identified in the regional FCM, but were found in the manually coded FCMs gener-

ated with smaller spatial frames. These results are indicative of a previously expressed sentiment from stakeholders indicating a lack of jurisdictional power as a major barrier of regional stormwater management [16]. With this co-developed knowledge, policy-makers, stormwater professionals, researchers, and educators need to recognize how spatially framing a stormwater problem influences the perceived range of possible problems, barriers, and solutions through spatial cognitive filtering of the system's boundaries.

*4.3. Limitations*

The results of this work are limited by the semi-qualitative nature of fuzzy cognitive mapping via subjective variable aggregation [6]. Variable aggregation can heavily influence outcomes even when using robust methods to reduce subjectiveness (e.g., dual coding). In addition, the regional, manually coded, and spatially filtered FCMs are not exact representations of the stakeholders' perceptions of stormwater management at a specific spatial scale and instead represent an approximate social cognitive model [6]. For example, while developing the manually coded FCMs, stakeholders did not explicitly define FCM variables or the connection weights between variables. Instead, the research team interpreted the variables and connections, and the connection weights were set only to maintain directional relationships (−1 or 1). However, eliciting FCM variables and connection weights from the survey with Likert scale responses, similar to other studies, was not an appropriate choice at this point because we did not want to limit stakeholders to predefined variables [3,6]. We also acknowledge there are disadvantages to only using the minimum and maximum activation values when comparing model results. One drawback is that we only compared the most exaggerated results. Another disadvantage is that we did not consider stakeholders' perceptions of how they would have activated the various mitigation strategies.

The work is also limited because we had a homogeneous and limited sample size of stakeholders and lacked an equivalent number of stakeholders from industry and NGOs as from government and academia. We also did not include other community members not directly involved with stormwater management, planning, or policy development. These additional stakeholders have perspectives of stormwater management that are essential to co-design future systems that can democratize decision-making [58,59]. In future studies, stakeholder perceptions of flood risk could be examined through the inclusion of communities where flooding occurs and does not occur.

## 5. Conclusions

The importance of defining spatial scale in environmental and ecological modeling is well known [12,52–55,60]. However, fuzzy cognitive models of these same SES often ignore spatial scale. When implicitly set, spatial scale adds additional complexity into the web of socio-environmental systems that form cognitive models [12]. The added complexity comes from the effects of the spatial frame used to develop the FCM. These effects filter the types of variables and relationships considered during modeling and impact model results. However, FCMs developed by different stakeholder groups may have the same effects. That is why explicitly setting spatial scale is essential when engaging diverse stakeholders when developing FCMs, and why it is vital to identify shared language to describe the problems, barriers, and solutions of the modeled system when working with diverse stakeholder groups.

In our study of the Triangle Region, North Carolina, we demonstrated that a regional FCM of stormwater management does not fully represent stakeholder perceptions of household-scale stormwater problems. Similarly, FCMs developed at household and neighborhood scales do not accurately represent stakeholder perceptions at regional spatial scales. While there is overlap between FCMs developed with varying spatial frames, FCMs should not be repurposed to address issues at spatial scales outside of the scale they were developed. More research is needed to understand how spatially framing problems impact the results of studies and how scaling can intentionally or unintentionally leave elements

of a SES hidden. Additionally, more research is needed to explore scaling relationships between variables within an FCM and to identify how relational weights are affected by scale.

We also identified that Academic and Governmental stakeholders use a wide range of language to describe stormwater problems. We recommend the development of a shared vocabulary for use in future modeling workshops. The results from the FCM can be used as a starting point that stakeholders can fine-tune. Additionally, stakeholders' descriptions of the spatial characteristics of stormwater problem areas were generally in agreement with the spatial data sets used for comparison. The spatial characteristics of the stakeholder-identified locations included areas where streams converge in developed areas within the floodplain where development occurred before current stormwater control regulations were introduced. These areas also all have upstream contributing areas with greater than 50% developed land cover.

Additionally, stakeholders' perceptions of the spatial scale generally matched the actual upstream contributing area of the problem. The drivers of the stormwater problems occurred mainly at the neighborhood or city scales, and impacts of the problems were mostly identified at neighborhood levels. However, a better approach to define scale should be used in future studies to remove ambiguity between neighboring hierarchical scales.

New geospatial participatory modeling methods should be developed by exploring the coupling of FCMs and spatially explicit data. These include improving the proposed coupling methodology and developing geospatial FCMs. For example, we recommend the development of a dynamic probabilistic FCM to map multi-scale SES over time. These models can be developed as part of a serious game that allows a broader range of stakeholders to provide ongoing input into the FCM. These geospatial fuzzy cognitive models would allow stakeholders to draw spatially explicit 3-dimensional graphs on a map representing how landscape elements are related at various scales. The results from a geospatial FCM may provide valuable insights into how stakeholders perceive how their communities and environments are connected. Geospatial models of socio-environmental systems can also be used with stakeholders to examine how well perceptions of space align with physical models [11,45,61,62].

In conclusion, we developed a technique to couple a regional FCM of stormwater management with a spatially explicit survey identifying stormwater problems in the Triangle, NC. This technique provides a pathway for future studies to further understand the effect of spatial scale on fuzzy cognitive models. Future studies may explore if patterns of variable and relationship selection exist across spatial scales, providing researchers and policy-makers guidelines for selecting appropriate spatial scales when working with diverse stakeholders [52]. Stakeholder engagement through participatory modeling will continue to be essential to finding sustainable solutions to complex socio-environmental issues of stormwater management. However, to create sustainable outcomes for complex socio-environmental issues surrounding stormwater management, we must continue to explore how stakeholders perceive and interact with spatial scale [12,54].

**Author Contributions:** Conceptualization, C.T.W., H.M., T.K.B., K.F., O.P. and R.K.M.; methodology, C.T.W.; software, C.T.W.; validation, C.T.W., H.M. and T.K.B.; formal analysis, C.T.W.; investigation, C.T.W. and O.P.; resources, R.K.M. and H.M.; data curation, C.T.W. and T.K.B.; writing—original draft preparation, C.T.W.; writing—review and editing, T.K.B., K.F., J.V., H.M., R.K.M. and O.P; visualization, C.T.W.; supervision, R.K.M. and H.M.; project administration, C.T.W. and O.P.; funding acquisition, R.K.M.,O.P., H.M., T.K.B. and K.F.; All authors have read and agreed to the published version of the manuscript.

**Funding:** This research was funded by the National Science Foundation grant number 1737563.

**Institutional Review Board Statement:** The study was conducted according to the guidelines of the Declaration of Helsinki, and approved by the Institutional Review Board (or Ethics Committee) of North Carolina State University (protocol code 12141, 5 October 2018).

**Informed Consent Statement:** Informed consent was obtained from all subjects involved in the study.

**Data Availability Statement:** Publicly available datasets were analyzed in this study. This data can be found here: https://drive.google.com/drive/folders/1uurw4oi179pF5YOkSY1En-YUMAb5P6 zx?usp=sharing (accessed on 26 August 2021).

**Acknowledgments:** The authors thank the members of the Research Coordination Network (RCN) from across the Research Triangle Region of North Carolina. The RCN consists of four academic institutions (North Carolina State University, the University of North Carolina at Chapel Hill, Duke University, and North Carolina Central University), seven municipalities (City of Raleigh, City of Durham, Town of Chapel Hill, Town of Cary, Wake County, Durham County, and Orange County), and NGOs (South Atlantic Landscape Conservation Cooperative, Water Resources Research Institute). The authors also thank Megan Skrip of the Center for Geospatial Analytics for extensive edits and suggestions that improved the manuscript.

**Conflicts of Interest:** The authors declare no conflict of interest.

## Appendix A. Regional Fuzzy Cognitive Maps

*Appendix A.1. FCM Variable Contingency by Stakeholder Group*

**Table A1.** When comparing academic and government stakeholder FCM variables that were mentioned more than once using Pearson's Chi-Square Test, we find that there is a significant relationship ($\alpha < 0.05$) between the stakeholder group and variables selection ($p = 0.019$).

| FCM Variable Contingency Table | | | |
|---|---|---|---|
| **Variable** | **Academic (n = 7)** | **Government (n = 10)** | **Total** |
| rainfall | 6.0 | 5.0 | 11 |
| flooding | 5.0 | 7.0 | 12 |
| impervious surfaces | 4.0 | 8.0 | 12 |
| climate change | 4.0 | 2.0 | 6 |
| retention/detention capacity | 3.0 | 3.0 | 6 |
| water quality | 3.0 | 3.0 | 6 |
| political will | 3.0 | 2.0 | 5 |
| runoff | 2.0 | 5.0 | 7 |
| population | 2.0 | 2.0 | 4 |
| stormwater management quality | 2.0 | NA | 2 |
| stormwater management regulations | 2.0 | NA | 2 |
| urban development with specific land uses | NA | 7.0 | 7 |
| property damage | NA | 5.0 | 5 |
| open space | NA | 4.0 | 4 |
| canopy | NA | 3.0 | 3 |
| development restrictions | NA | 3.0 | 3 |
| erosion | NA | 3.0 | 3 |
| natural areas | NA | 3.0 | 3 |
| stream/ecosystem health | NA | 3.0 | 3 |
| traffic | NA | 3.0 | 3 |
| development in floodplain | NA | 2.0 | 2 |
| green infrastructure | NA | 2.0 | 2 |
| highway commercial development | NA | 2.0 | 2 |
| local economy | NA | 2.0 | 2 |
| scm | NA | 2.0 | 2 |
| streamflow | NA | 2.0 | 2 |
| tax base | NA | 2.0 | 2 |
| **Total** | 36.0 | 85.0 | 121 |

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
