# Peer review of "Spatially Explicit Fuzzy Cognitive Mapping for Participatory Modeling of Stormwater Management"

_land, doi:10.3390/land10111114_

Round 1

Reviewer 1 Report

The paper presents an interesting methodology to couple spatially explicit surveys and Fuzzy cognitive Maps (FCMs) in order to understand the role of spatial scale in stakeholders’ conceptualization of urban stormwater related problems, with particular focus on 3 cluster areas In North Carolina (USA). Overall, the manuscript provides insight into both academics and governments’ understanding of stormwater management at regional scale. I suggest the acceptance of the manuscript with minor revisions.

In fact, in the Introduction the focus of FCMs is well deepened. However, to better situate the research, more literature review should be added on the need to build a shared understanding of sustainability problems and how the spatial scale affects the FCMs variables and processes. Moreover, the whole manuscript is biased in favor of the methodology section. I suggest broadening the Discussion of the results by comparing the outcomes with the available literature. Furthermore, the figures should be contextualized with smaller scale maps. For instance the collocation of the Triangle region in the USA map should be visible in Figure 1. Finally, I recommend to expand the presentation of the main findings of the research in the Conclusion section.

See specific comments below for further detail.

In Chapter 1, I suggest to move the description of the Triangle region to the end of the section, in order to introduce the reader to the general topic and not the specific case study.

Chapter 2 is quite difficult to follow. Maybe a flowchart or an introductory paragraph that explains the steps taken by the research could be of help for the reader.

In Chapter 2.2 it is not clear if the stakeholders of the second workshops are the same of the first workshops. Also, please provide a brief description of the spatial survey application “Survey123” for ArcGIS.

In Chapter 2.3.1 please explain the meaning of A and B of Equation (1).

In Chapter 2.3.2 please explain the meaning of Ec  of Equation (2).

In Chapter 2.5 the disadvantages of the activation limited to the "a lot" variables should be acknowledged.

In the caption of Table 1, I believe that (p < 0.05) should be replaced with (p > 0.05).

In Chapter 3, the differences between the upper and the lower graphs of Figure 8 should be clarified.

In Chapter 3.3 the locations and landscape elements (e.g. Little Creek) described in the text should be reported in the related Figures or avoided. Also, the figures for Cluster 3 are missing without explanation.

Reviewer 2 Report

The manuscript topic is one that should engender interest.  However, there are some significant deficiencies in the methodology; these deficiencies impact the application of the proposed approach rather than impact the proof of concept presented.  One of these deficiencies is the concept of risk and how that risk changes if the community does not encounter flood events.  A second deficiency is that mitigation works at one location in the watershed will impact the flood risk at other locations in the watershed.  How was the interconnectivity of local problems across the watershed considered; the watershed interconnectivity seemed to be considered only for regional issues.
